

# Nonlinearity of transparent SNS weak links decreases sharply with length

Valla Fatemi[1]*, Pavel D. Kurilovich[2], Anton R. Akhmerov[3] and Bernard van Heck[4]

**1** Cornell University, Ithaca, NY, USA
**2** Yale University, New Haven, CT, USA
**3** TU Delft, Delft, The Netherlands
**4** Sapienza Università di Roma, Rome, Italy

* vf82@cornell.edu

## Abstract

Superconductor-normal material-superconductor (SNS) junctions are being integrated into microwave circuits for fundamental and applied research goals. The short junction limit is a common simplifying assumption for experiments with SNS junctions, but this limit constrains how small the nonlinearity of the microwave circuit can be. Here, we show that a finite length of the weak link strongly suppresses the nonlinearity compared to its zero-length limit – the suppression can be up to a factor of ten even when the length remains shorter than the induced coherence length. We tie this behavior to the nonanalytic dependence of nonlinearity on length, which the critical current does not exhibit. Further, we identify additional experimentally observable consequences of nonzero length, and we conjecture that anharmonicity is bounded between zero and a maximally negative value for any non-interacting Josephson junction in the presence of time-reversal symmetry. We promote SNS junction length as a useful parameter for designing weakly nonlinear microwave circuits.

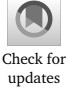

# 1 Introduction

Superconductor-semiconductor Josephson weak links are being incorporated to superconducting quantum circuits to provide parametric control through the field effect [1–7] and microscopic internal degrees of freedom with long-range interconnectivity [8–17]. The standard approach for modeling such devices, motivated by a desire for simplicity and by the fact that junctions are often made as short as possible, is the use of the short junction formula for the ground state energy-phase relation [1, 3, 4]:

$$U_{\text{short}}(\varphi) = -E_0 \sqrt{1 - \tau_{\text{eff}} \sin^2(\varphi/2)}, \tag{1}$$

where $E_0$ is an energy scale and $\tau_{\text{eff}}$ is an effective transparency. This approach is often justified by appealing to the idea that nonzero length introduces corrections small in $l = L/\xi$, where $L$ is the junction length and $\xi$ is induced coherence length.

However, typical superconductor-semiconductor weak links, particularly in devices based on the InAs/Al platform [18, 19], have lengths comparable to $\xi$. Therefore, finite-length corrections might be important at a non-perturbative level to describe experiments with microwave circuits incorporating SNS weak links [20]. A nonzero dwell time in dot-like weak links was found to strongly impact the energy-phase relations and microwave properties of the ground state [20–22]. As far as we know, however, a direct inspection of the impact of junction length was not conducted. We promote that the dwell time can present a unifying picture between the two cases. For a dot, the dwell time is given by the inverse of the tunneling rate between the dot and the reservoirs. For a nonzero length junction, if we take $\xi$ as proportional to the Fermi velocity $v_F$, we can identify the dwell time as the time of flight $L/\xi \propto L/v_F$ between the reservoirs.

Here, we inspect two basic models for single-channel SNS junctions that deviate from the usual short junction limit: a short resonant level model [21, 23] and a ballistic finite length model [22]. We show how both models display reductions in the anharmonicity when the continuum contribution to the phase-dependent part of the ground-state energy starts to play a role, including nonanalytic behavior as the dwell time in the weak link approaches zero. Crucially, even for $l \lesssim 1$, while the junction hosts only single Andreev level, the anharmonicity can change by a factor of 10 from that of an idealized short junction ($l = 0$), whereas the critical current decays only by a factor of 2. Therefore we promote this 'shortish' regime of $0 < l \lesssim 1$ as distinct from both the short and long junction regimes. Incidental experimental evidence for this reduced anharmonicity is already present in the literature, both in measured inductance phase relations of SNS junctions [22] as well as anharmonicity of SNS gatemons [3] (see Appendix A). We conclude with calculations that consider ways to identify the effect of the finite length in a shortish junction. We advocate for care both in choice of models to explain experimental data and in including the continuum contribution to the SNS weak links, particularly in superconducting microwave circuits which will be particularly sensitive to these effects [22, 24, 25]. Finally, we remark on possible utility of the low anharmonicity in shortish, transparent SNS junctions.

## 2 Main results

### 2.1 Introductory information

Given an energy phase relation $U(\varphi)$, we define the polynomial expansion about the minimum as

$$U(\varphi) = \sum_{n=0}^{\infty} c_n \varphi^n. \tag{2}$$

We focus on the ground state of weak links without charging energy and with time reversal symmetry, so that the minimum is always at $\varphi = 0$ and symmetric, resulting in $c_n = 0$ for odd $n$. The second order coefficient $c_2$ is proportional to the inverse linear inductance.

Consider a junction shunted by a capacitor which provides a charging energy $E_C$, resulting in the Hamiltonian of the resulting mode being $H = U(\hat{\varphi}) + 4E_C \hat{n}^2$. The characteristic impedance of the mode can be said to be proportional to $\sqrt{E_C/c_2}$, and is well-defined when it is small, achieving a limit where the device may be called a transmon. The transition frequency from the ground to the first excited state is then given by $hf_{01} \propto \sqrt{c_2 E_C}$, and the anharmonicity relative to the charging energy is

$$\alpha_r = (hf_{12} - hf_{01})/E_C \tag{3}$$

$$= 12c_4/c_2, \tag{4}$$

where the second equality is true in the limit of phase fluctuations tending to zero, in which the $\phi^4$ term in the expansion (2) can be treated perturbatively. We remark that this ratio also characterizes the fourth-order nonlinearity used in parametrically pumped circuits [26]. Throughout this manuscript, we will assume this small-fluctuation limit for simplicity, except in Section 3.2 when we consider realistic parameters for transmon qubits.

For a tunnel junction with $U(\varphi) = -E_J \cos \varphi$, $\alpha_r = -1$ because $c_2 = \frac{1}{2}$ and $c_4 = -\frac{1}{24}$. Reference [1] recently pointed out that $\alpha_r = -\frac{1}{4}$ for a short junction-like energy phase relation (1) at transparency of $\tau_{\text{eff}} = 1$. Indeed, Eq. (1) has $c_2 = \frac{E_0}{8}$ and $c_4 = -\frac{E_0}{384}$. The short junction model can smoothly interpolate to the tunnel junction by varying $\tau_{\text{eff}} \to 0$:

$$\alpha_r = -1 + \frac{3}{4}\tau_{\text{eff}}. \tag{5}$$

Finally, before continuing to our results, we remark on the length of a junction $L$, which we will consider in ratio to the induced coherence length $\xi$: $l = L/\xi$. In the long-length limit $l \to \infty$, the current-phase relation of a ballistic device approaches a sawtooth function [27], and therefore the junction is approximately harmonic. Similar results appear to hold for the diffusive case [28]. In this work, we will be mostly concerned with the regime $l \lesssim 1$.

## 2.2 Zero-length resonant level

The resonant level model of Beenakker and van Houten [23] smoothly interpolates between an ideal short junction and a junction with a large dwell time by varying the coupling between the resonant level and the superconducting leads (see schematic in Fig. 1(a)); in between these limits, we will find a reduced anharmonicity. This model captures the effect of a finite dwell time in the junction, but it remains "short" in the sense that it always contains a single Andreev bound state, as a consequence of taking the level spacing of the normal region to be $\gg \Delta$. The model is formulated in Appendix B.2. Even though it does not explicitly contain a length parameter, the dwell time can be used as a common parameter for comparisons to a junction of finite length.

The model is defined by a single fermionic level with level offset $\epsilon_0$ coupled to two superconductors with gap $\Delta$ by tunneling rates $\Gamma_L$ and $\Gamma_R$. The dwell time is $\propto \Delta/\Gamma_\Sigma$ with $\Gamma_\Sigma = \Gamma_L + \Gamma_R$, and the critical current on resonance ($\Gamma_L = \Gamma_R, \epsilon_0 = 0$) scales with the dwell time as $I_c = I_{c0}/(1 + \Delta/\Gamma_\Sigma)$, where $I_{c0}$ is the critical current at zero dwell time. The short junction limit is given by $\Gamma_\Sigma \gg \Delta, \epsilon_0$, while the dot-like resonant level is given by $\Gamma_\Sigma, \epsilon_0 \ll \Delta$. In both these limits, the ground state is described by equation (1) [29], and all of the ground state energy-phase relation comes from the sub-gap Andreev bound state.

In between these limits, the energy phase relation is different and has contributions from both the bound state and the continuum [21–23]. The deviations from Eq. (1) are expected to be strongest in the vicinity of $\Gamma_\Sigma \sim \Delta$ and $\tau \to 1$. We explicitly evaluate the ratio $\alpha_r$ for this model for a large range of $\Gamma_\Sigma$ and $\tau_{\text{eff}}$, as shown in Fig. 1(c) and Fig. 6(c). The limit $\tau_{\text{eff}} \to 0$ (black) reproduces the tunnel junction case, while the limits $\Gamma_\Sigma \gg \Delta$ and $\Gamma_\Sigma \ll \Delta$ reproduce the physics of Eq. (1). Varying the dwell time from zero, we find nonanalytic behavior (see later sections), and when $\Gamma_\Sigma \sim \Delta$ and $\tau_{\text{eff}} \sim 1$, the relative anharmonicity is reduced to as low as 0.1 of the tunnel junction case (2.5× weaker than the lowest of the short junction model). Notably, the deviations are relevant over several orders of magnitude of $\Gamma_\Sigma$.

## 2.3 Nonzero-length ballistic weak link

Most experiments are in the regime $l \sim 1$, so the resonant level model will miss the impact of the normal region level spacing being comparable to $\Delta$. To capture this effect, we employ a single-channel finite length model described in the supplement of Ref. [22] and shown schematically in Fig. 1(b) (with full details in Appendix B.3). The model consists of a ballistic normal region contacting superconductors at interfaces with reflection coefficients $R_L$ and $R_R$. Resonant effects occur when the reflection coefficients are equal, and when $R_L = R_R = 0$ we find the same scaling with dwell time as the fully transparent resonant level model [30,31]:

$$I_c = I_{c,\text{short}} \frac{1}{1+l} \, . \tag{6}$$

Therefore, the critical current of a ballistic SNS junction does not exhibit nonanalytic behavior with length, unlike a diffusive junction [32].

Figure 1(d) shows the relative anharmonicity for this model as a function of $l$ for different $R_L$ from 0 to 1 (keeping $R_R = 0$). In this situation we can assign an effective transparency $\tau_{\text{eff}} = 1 - R_L$. For $l = 0$, the short junction model is reproduced; then, increasing the length

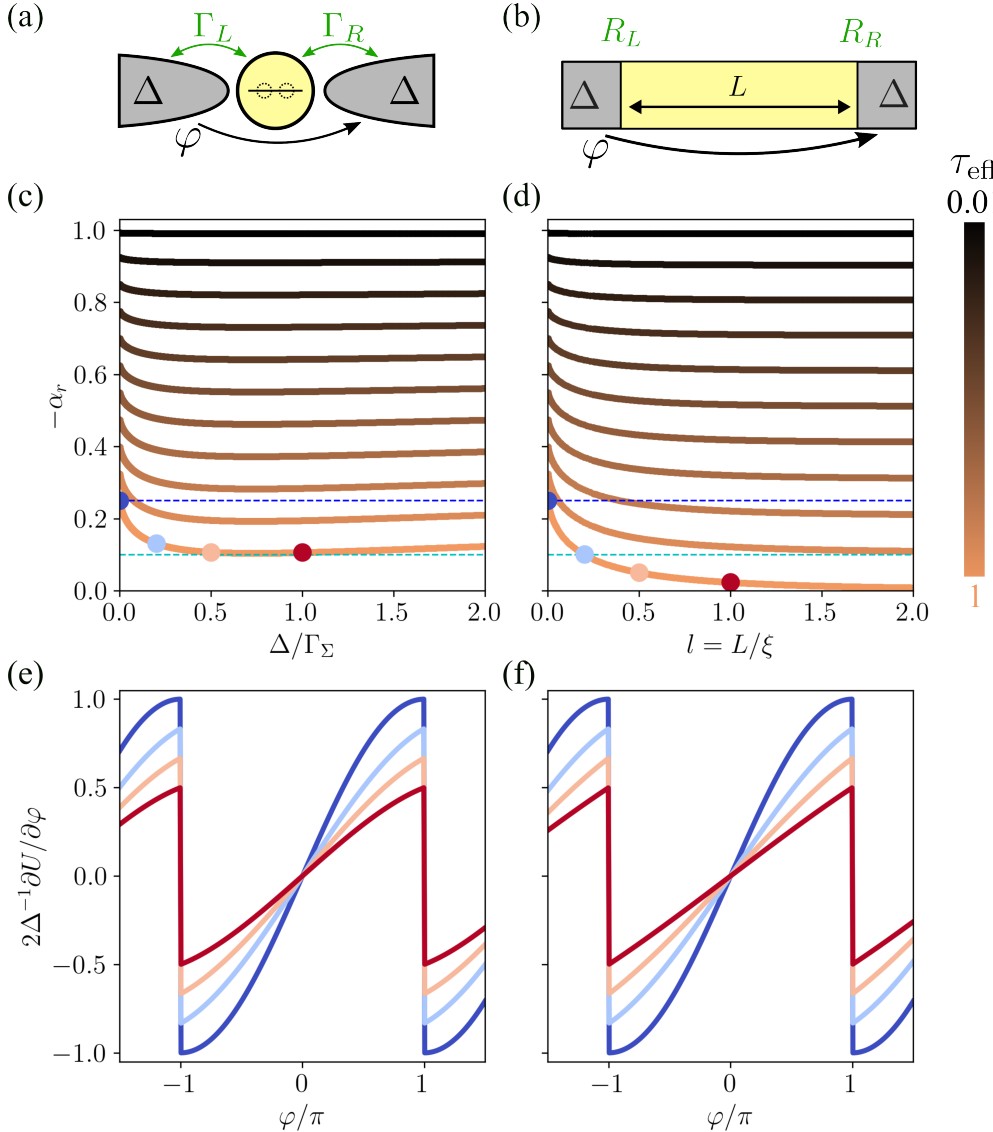

Figure 1: **SNS transmon anharmonicity in small phase fluctuation limit.** (a) Schematic of the short resonant level model (from [22]). (b) Schematic of the finite length single-channel model. (c-d) Relative anharmonicity $-\alpha_r$ for both models, for evenly-spaced transparencies, with the lower axis the parameter proportional to the dwell time in the circuit. (c) The resonant level model, as a function of the inverse total coupling strength for several different effective transparencies $\tau_{\text{eff}}$. (d) Finite length model, as a function of length for different $R_L = 1 - \tau_{\text{eff}}$, fixing $R_R = 0$. The blue dashed line is at 0.25 (lower limit of short junction formula), cyan dashed line is at 0.1 (lower limit of resonant level model). The same figure but with logarithmic scale on the lower axis is given in Appendix B.4. (e) and (f) Show the current-phase relations at the designated points in (c) and (d), respectively. The long junction has a well-developed sawtooth like shape at $l = 1$.

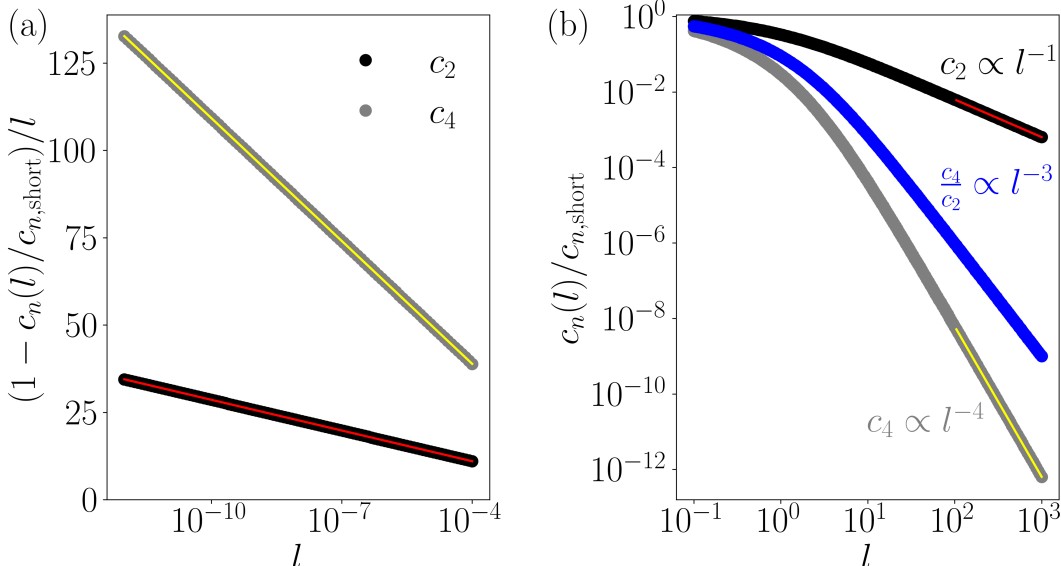

Figure 2: **Trends with length approaching the short and long limits** (a) The short-length variation of $c_2$ and $c_4$ scaled to emphasize the nonanalytic behavior. Solid lines are fits to Eq. (7), with $b_n$ the only fit parameter and $a_2 = 1/2\pi = 12a_4$. (b) The long-length variation of $c_2$ and $c_4$ and their ratio. Solid lines are fits to a power law.

from 0, the anharmonicity drops rapidly as long as $\tau_{\text{eff}} \gtrsim 0.5$ – indeed showing nonanalytic behavior. When $R_R > 0$, the lowest anharmonicities occur for the resonant condition $R_R = R_L$ and still allow anharmonicities well below 0.1.

## 2.4  Nonanalytic behavior at small dwell time

We find that both models exhibit nonanalytic dependence with dwell time in the first two polynomial coefficients, following:

$$c_n = c_{n,\text{short}} - a_n l \ln\left(\frac{b_n}{l}\right), \qquad n = 2, 4, \qquad l \ll 1 \tag{7}$$

(for the dot, we would substitute $l$ with $\Delta/\Gamma$). This is somewhat remarkable because the critical current, which follows Eq. (6), does not exhibit such a nonanalytic behavior in these models [30].

We inspect $c_2$ and $c_4$ at $R_L = R_R = 0$ for the nonzero-length model in Fig. 2(a), finding that they follow Eq. (7) with differing coefficients: $a_2 = 1/2\pi$, $b_2 \approx 0.5615$, $a_4 = a_2/12$, $b_4 \approx 0.2066$. We derived the $a_n$ coefficients exactly (see Appendix C), and the $b_n$ coefficients were extracted by fitting to the computed quantities in Figure 2(a). Because the relative changes of $c_4$ and $c_2$ are different, the anharmonicity also exhibits a nonanalytic length dependence, visible in Fig. 1. Indeed, for $\tau_{\text{eff}} = 1$ we find

$$\alpha_r \approx -12 \frac{\frac{1}{384} - \frac{1}{12}\frac{l}{2\pi}\ln\left(\frac{b_4}{l}\right)}{\frac{1}{8} - \frac{l}{2\pi}\ln\left(\frac{b_2}{l}\right)} \approx -\frac{1}{4}\left[1 - \frac{24l}{2\pi}\ln\left(\frac{b_\alpha}{l}\right)\right], \qquad b_\alpha = b_4^{4/3}b_2^{-1/3} \approx 0.148. \tag{8}$$

The resonant level model also exhibits nonanalytic behavior in the dwell time, which is given by $\Delta/\Gamma_\Sigma$ rather than $l$. This model has identical $a_2$ and $a_4$, but distinct $b_2$ and $b_4$ (see Appendix C).

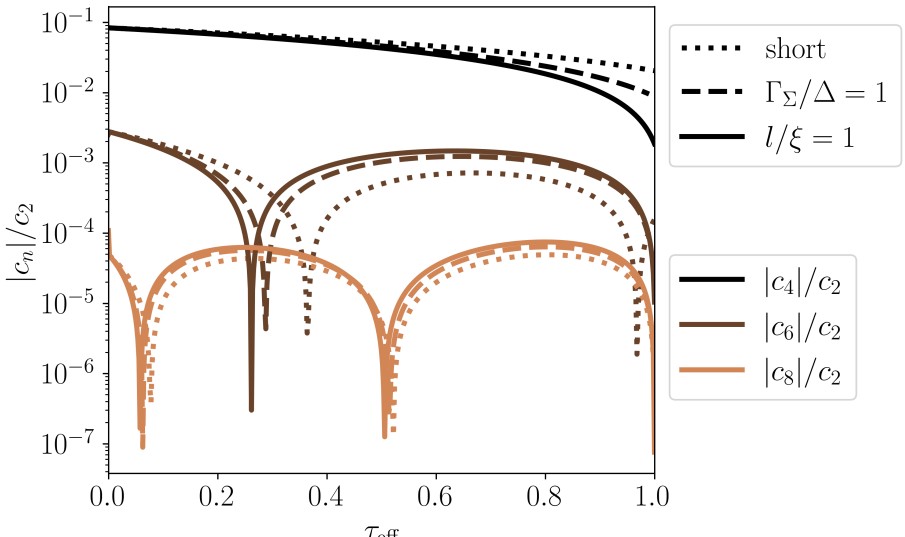

Figure 3: **Higher Order Polynomial Coefficients** Fourth, sixth, and eighth order polynomial coefficients of the potential, normalized to $c_2$, as a function of $\tau_{\text{eff}}$ for the short junction (dotted line), resonant level with $\Gamma_\Sigma = \Delta$ (dashed lines), and finite length junction with $l = 1$ (solid lines). Notably, the sixth order coefficients are larger for the models with nonzero dwell time, and effective transparency at which $c_6 = 0$ shifts to lower transparency. Note that the dips are associated with sign changes.

Finally, we also comment on the long-length limit $l \gg 1$, in which the coefficients exhibit a power-law decay with length [Fig. 2(b)]: $c_2 \propto l^{-1}$, consistent with results on supercurrent previously published [33,34], and $c_4 \propto l^{-4}$. Therefore we find $\alpha_r \propto l^{-3}$. Although we are not aware of this particular scaling having been described previously, the rapid decay is consistent with the long-junction limit exhibiting a sawtooth-like current-phase relation [33,34]. Notably, the scaling of anharmonicity with length is one order faster than the scaling of anharmonicity achieved by arraying tunnel junctions [35, 36].

To summarize, we find that the key reactive (e.g., microwave) properties of transparent Josephson weak links follow a nonanalytic length dependence such that the junction nonlinearity can be suppressed by a factor of 10 even at shortish lengths (when comparing to zero length).

## 2.5 Evidence for guarantee of bounded, negative-definite anharmonicity

Our results show that the anharmonicity of SNS junctions can decay sharply with length and approaches zero rapidly. It then becomes natural to ask whether the anharmonicity must be negative or whether it can change sign. To study this question, we turn to the properties of time-reversal symmetric, noninteracting SNS junctions, with s-wave superconductors.[1] These exhibit a minimum in their energy phase relations at $\varphi = 0$ – it was proven before that $c_2 \geq 0$ [38]. Here we suggest additionally that $c_4$ may also have guarantees of sign (negativity) and scale.

---

[1]Josephson junctions with d-wave superconductors can behave differently [37].

The two scattering models we consider have a common structure, allowing us to write down

$$\alpha_r = -1 + 3 \frac{\int_{-\infty}^{\infty} |\bar{\Lambda}(i\epsilon)|^{-2} \, d\epsilon}{\int_{-\infty}^{\infty} |\bar{\Lambda}(i\epsilon)|^{-1} \, d\epsilon}, \tag{9}$$

where $\bar{\Lambda}(i\epsilon)$ is energy-dependent part of the (model-dependent) scattering amplitude at $\varphi = 0$ with a normalization. These models have the property

$$-1 \leq \alpha_r \leq 0. \tag{10}$$

The upper limit $\alpha_r \to 0$ is approached only as $l \to \infty$ in the finite length model, while the resonant model has an upper bound $\alpha_r \approx -0.103$. See Appendix D.1 for details.

This motivated us to inspect more general cases, so we checked millions of random Hamiltonian matrices satisfying the constraints of an SNS junction at $\varphi = 0$ – these also respect the bounds $-1 \leq \alpha_r \leq 0$ (details in Appendix D.2). Combined with the guarantee on $c_2$, this would ensure that low-impedance SNS transmons will always have negative anharmonicity that is never stronger than a tunnel junction. Perhaps the approach used by Titov et al. [38] could be extended to fourth order to assess if these observations are analytically general for SNS weak links with these constraints.

# 3 Distinguishing short from shortish junctions

## 3.1 Higher components of the polynomial expansion

Inspection of the polynomial coefficients is particularly helpful for situations where the junction has a small participation in a resonant mode, so that these coefficients will directly relate to varying orders of nonlinear processes [35, 36]. In particular, the negativity of the fourth-order nonlinearity $c_4$ does not apply to higher order terms. We find that the sextic term ($c_6$) and others pass through zero at intermediate transparency, and that this zero depends sensitively on the model and its parameters - see Figure 3. The higher order polynomial coefficients will contribute, for example, higher order contributions to Stark shifts in strongly driven circuits, which may be desirable to tune larger or smaller depending on the application. Also, zeros of these different coefficients can serve to constrain models for the weak link energy-phase relation.

## 3.2 Effect of nonzero impedance

For cases in which the junction has a high participation in the circuit and the impedance is not near zero, typical of SNS transmons, we can instead consider the higher anharmonicities. The hierarchy of anharmonicities will be sensitive to all aspects of the junction energy-phase relation and can serve as a useful way to diagnose the overall energy-phase relation [25]. We define

$$\alpha_{ijk}/h = f_{jk} - f_{ij}, \tag{11}$$

as the absolute anharmonicity between the $jk$ and $ij$ transitions, where $\alpha_{012}$ corresponds to the previously inspected anharmonicity. We remark that the offset-charge dispersion may also hold useful diagnostic information, but in this work we avoid this issue by operating at resonance (where such dispersion is nominally quenched [3, 4, 39, 40]) and setting the offset charge to the charge parity degeneracy point so that the anharmonicity is always computed at the charge sweet spot [41].

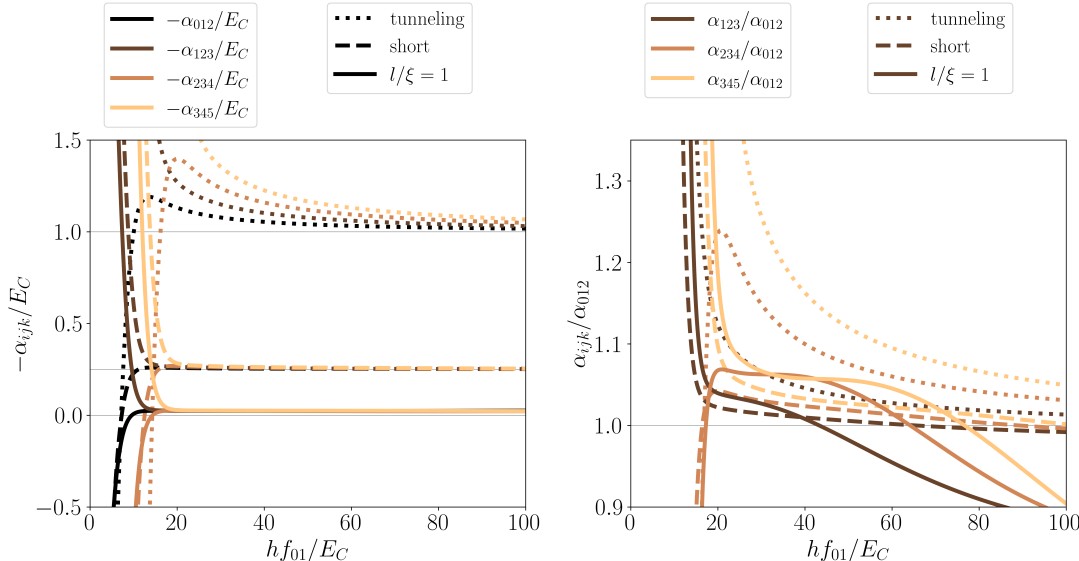

Figure 4: **Nonzero Impedance and Higher Level Anharmonicities** The lowest four anharmonicities for three situations: tunneling junction, short junction, and finite length junction at $l = 1$ and $R_1 = 0.01$ ($R_2 = 0$). The offset charge has been set to the degeneracy point. (a) The raw anharmonicities relative to $E_C$, showing the hierarchy at low impedance (high frequency). (b) The higher anharmonicities relative to the lowest anharmonicity. We only show the range of values where the lowest anharmonicity remains negative $\alpha_{012} < 0$. From this perspective, the finite length model appears more similar to the tunneling junction than the conventional short junction.

In Figure 4(a), we show calculations of the first four anharmonicities as a function of a multiplicative factor on the energy-phase relation, keeping the effective transparencies near 1 for the short and finite length junctions. We plot them as a function of the ratio of the lowest transition frequency and the charging energy, $hf_{01}/E_C$, because $E_C$ is usually well approximated by simulation of the circuit design and $f_{01}$ is directly observed experimentally. This ratio, for high values, is roughly proportional to the inverse of the characteristic impedance. On an absolute scale (for fixed $E_C$), the anharmonicities of highly transparent junctions are more stable to variation in the impedance than tunnel junctions, before diverging quickly below around $hf_{01}/E_C \lesssim 13$. Note also how the anharmonicities converge more quickly as the impedance is lowered for the long junction; this may be related to the impact of higher order nonlinearities also scaling with the transparency.

However, by inspecting the anharmonicities relative to each other, we can gain more nuanced information. We find that in this respect the finite length junction behaves more similarly to a tunnel junction than the short junction for a typical transmon with $hf_{01}/E_C \sim 15 - 40$ (Fig. 4(b)), although the anharmonicities relative to $E_C$ are drastically different (Fig. 4(a)). For even lower impedance transmons, the intermediate length junction has more dramatic deviations, with higher anharmonicities dipping below the first anharmonicity. Therefore, we propose that inspection of these quantities together can provide more detailed information to constrain models of experimental devices.

# 4 Discussion

## 4.1 Practical implications for basic science investigations

Most experimental devices are in the vicinity of $l \sim 1$. As a result, it is unlikely that the short junction formula is applicable. We suspect that previous work missed our observations because small quartic terms are difficult to see on the quadratic background, and the decay of critical current with length is comparatively slow, resulting in the idea that any difference may be thought to be irrelevant. In fact, as we have seen here, the deviations from Eq. (1) around $\varphi = 0$ are strong enough to have a large effect on the anharmonicity. Analysis of publicly available data from published experiments already reveals cases of $-\alpha_r < 0.25$ and even $< 0.1$ (see Appendix A). Finally, we note that lithographic resolution in academic labs is likely to remain relatively fixed while there are active efforts to incorporate superconductors with higher gaps (shorter coherence lengths) than aluminum onto super-semi devices [42–47]. This means that the short junction regime will become even harder to reach, thereby rendering the finite-length physics more relevant to future, higher gap devices.

How can we detect the effects of nonzero length when measurements indicate $\alpha_r < -0.25$? First, we observe that the short junction approaches the bound $\alpha_r = -0.25$ linearly in the transparency, per Eq. (5). Therefore, given that many devices exhibit significant mesoscopic fluctuations from disorder, it should be unlikely that the anharmonicity remains close to the short junction bound over a wide range of gate voltage configurations. Moreover, when the transparency is near unity, we would expect vanishing charge dispersion [39, 40], inspection of which alongside anharmonicity is not always possible or reported.

However, as we can see in Fig. 1(d), finite-length models allow for a much larger phase space of parameters yielding an anharmonicity near the extreme of the short junction model of Eq. (1). These cases have the additional feature of not requiring vanishing charge dispersion. Thus, finite-length models can account for the typically observed anharmonicities in the $-0.25E_C$ to $-0.5E_C$ range. It will be interesting to extend more sophisticated multi-channel SNS models to the finite length regime to observe their behavior in this respect.

Finally, we remark that models for diffusive SNS do exhibit non-anlytic behavior of the critical current [32]. Understanding the microwave properties of diffusive models will also be important for experiments.

## 4.2 Possible impact of charging effects

It has previously been speculated that deviations from short junction physics could result from charging effects, rather than finite length effects [4]. Indeed, perturbative charging effects do impact the continuum dispersion [21], in addition to the parity stability of the weak link. As an initial foray to this, we inspected the anharmonicity of the resonant level model with perturbative charging effects [21]. For the range $U/\Gamma_\Sigma \leq 0.5$ ($U$ is the charging energy), we find that charging effects have less than a 3% effect on the anharmonicity of the ground state (see Appendix E). Understanding the impact of charging effects, or Coulomb interactions more generally, on anharmonicity for finite length junctions remains for future work.

## 4.3 Mystery of the wide junction

Recently, a wide SNS junction was reported to exhibit the ratio $c_4/c_2$ three orders of magnitude lower than the tunnel junction limit [48]. The length of the junction was $\sim 100\,\text{nm}$, typical for such devices. Inspecting our Fig. 1(d), perhaps the first order of magnitude of discrepancy could be explained by the combination of high transparency and finite length effects described here. We suggest that the reported data indicates additional physics leading

to low anharmonicity in wide SNS junctions. As one possibility, wide junctions have recently been proposed to host long quasiparticle trajectories travelling along the width of the normal region [49]. These long trajectories may be in the truly long junction limit such that their contribution to anharmonicity may be low, e.g., a wide junction may also appear to be longer than it seems because the median electron trajectory is longer than the shortest distance between the superconducting leads. As another possibility, the junctions in [48] may have enough scattering to be closer to a diffusive regime, which may have different behavior in the shortish reigme than the ballistic model discussed here [28]. We leave a detailed inspection of this to future work.

### 4.4   Prospective usefulness of low anharmonicity in a single junction

Although high anharmonicity is useful for conventional qubits (like transmon qubits) where single-photon control is crucial, low anharmonicity is useful in a wide variety of contexts requiring efficient handling under the influence of high amplitude driving. Examples include parametric amplifiers [35, 36, 50–53], parametric beamsplitters [54–56], driven-dissipative qubits [57, 58], and switches for signal routing [59, 60]. Typically, reducing anharmonicity from that of a tunnel junction is achieved by reducing the participation of any one junction on the circuit, often by arraying tunnel junctions in series; arraying scales the Kerr nonlinearity roughly as an inverse square of the number of junctions [35, 36]. Other efforts focus on high kinetic inductance films which have much lower nonlinearity, often resulting in high pump power requirements causing thermal management issues [61–64]. We have shown here that a single intermediate-length, high-transparency SNS weak link can achieve reductions in anharmonicity comparable to arraying up to 10 tunnel junctions in series, offering an alternative approach for intermediate nonlinearity with small on-chip footprint that can be useful for a variety of device applications.

# Acknowledgments

V. F. acknowledges stimulating discussions with Sergey Frolov, Amrita Purkayastha, and Amritesh Sharma about experimental data which inspired this investigation as well as with David Feldstein Bofill, Zhenhai Sun, Svend Krøjer Møller, and Morten Kjaergaard. V. F. thanks Nicholas Frattini and Shyam Shankar for their correspondence on topics in the discussion section. P. D. K. acknowledges useful discussions with Charlotte Bøttcher, Thomas Connolly, and Elifnaz Önder. We thank Leonid Glazman, Sergey Frolov, Amrita Purkayastha, Amritesh Sharma, and Alex Levchenko for feedback and corrections to the manuscript.

**Funding information**   Research by V. F. and P. D. K. was sponsored by the Army Research Office and was accomplished under Grant Number W911NF-22-1-0053. The views and conclusions contained in this document are those of the authors and should not be interpreted as representing the official policies, either expressed or implied, of the Army Research Office or the U.S. Government. The U.S. Government is authorized to reproduce and distribute reprints for Government purposes notwithstanding any copyright notation herein.

**Data and code availability**   All data and code used in this manuscript are available online [65].

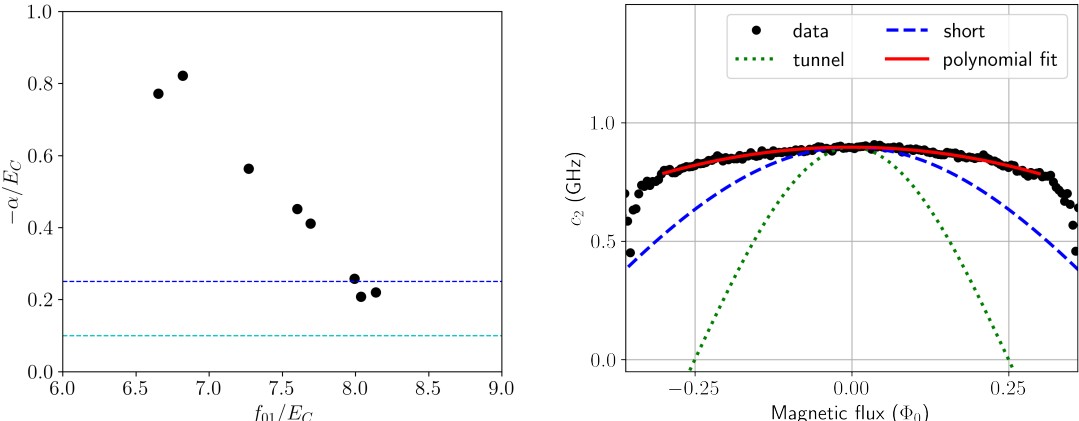

Figure 5: **Experimental evidence.** *Left:* Data from Figure 5 of Kringhøj, et al, [3], recast to show the anharmonicity explicitly. The anharmonicity goes below the limit allowed by short junction physics [1]. *Right:* Data from Figure S10 of Fatemi, et al., [22], recast as $c_2(\Phi)$, where $\Phi$ is a magnetic flux through a loop that phase biases a super-semi weak link (see reference for more details). The dispersion clearly has less curvature than either a tunnel junction (green dotted line) or perfectly transparent short junction (blue dashed line) with equivalent zero-flux $c_2$. The red line is a fit to a quadratic function to extract $c_4$, giving $-12c_4/c_2 = 0.07$.

**Author contributions** V. F. initiated and led the research. P. D. K. derived the scattering models used here and analytically obtained the prefactors for the logarithmic divergences. A. R. A. initiated and developed the random matrix approach described in Section 2.5 and Appendix D.2. B. vH. guided several aspects of the research and provided the code for calculating transmon energy levels for an arbitrary energy-phase relation. V. F. wrote the manuscript, which was reviewed by all authors.

# A Experimental support

We recast the publicly posted data for Figure 5 in Kringhøj, et al, [3] to Figure 5(left) in this manuscript. Near resonance (high transmon frequency), the anharmonicity dips down to a minimum of $0.21E_C$.

We also used the dispersive shift data from Figure S10 in Fatemi, et al, [22], which was converted to $c_2$ for Figure 5(right) here. A tunnel junction and short, transparent junction do not match the data at a qualitative level. Fitting the curvature of $c_2$ allows us to extract $c_4$, finding $-12c_4/c_2 = 0.07$, indeed below what is possible with the short junction formula or the resonant level formula. Although the resonant level formulation was used in that work, it was indeed unable to fully reconcile the inductance with its curvature.

# B Model details

We consider the adiabatic response of the weak link, which means that we consider the limit where resonant effects are not relevant. It has been argued that resonant effects dominantly affect the charge dispersion and not the anharmonicity [4].

## B.1 Generic aspects of both models

In both models, we consider a scattering amplitude $\Lambda(\epsilon, \varphi)$, which can be separated into energy-dependent and phase-dependent parts:

$$\Lambda(\epsilon, \varphi) = \Lambda_0(\epsilon) + \Lambda_\varphi(\varphi), \tag{B.1}$$

where $\epsilon$ is energy in units of $\Delta$ and the phase-dependent part is of the form

$$\Lambda_\varphi(\varphi) = v(1 - \cos(\varphi)), \tag{B.2}$$

with $v$ a model-dependent constant. Then, after defining $\bar{\Lambda} = \Lambda_0(\epsilon)/v$, can write the ground state energy as

$$U(\varphi) = -\frac{1}{2\pi} \int_{-\infty}^{\infty} \log\left(1 + \frac{1 - \cos(\varphi)}{\bar{\Lambda}(i\epsilon)}\right) d\epsilon. \tag{B.3}$$

The coefficients $c_n$ can be directly obtained by taking derivatives of the argument and then integrating:

$$c_n(\varphi) = -\frac{1}{2\pi n!} \int_{-\infty}^{\infty} \frac{\partial^n}{\partial \varphi^n} \log\left(1 + \frac{1 - \cos(\varphi)}{\bar{\Lambda}(i\epsilon)}\right) d\epsilon. \tag{B.4}$$

We give the full forms of $c_2$ and $c_4$ explicitly:

$$D = \bar{\Lambda}(i\epsilon) + 1 - \cos(\varphi), \tag{B.5}$$

$$c_2(\varphi) = -\frac{1}{4\pi} \int_{-\infty}^{\infty} \left[\frac{\cos(\varphi)}{D} - \frac{\sin^2(\varphi)}{D^2}\right] d\epsilon, \tag{B.6}$$

$$c_4(\varphi) = -\frac{1}{48\pi} \int_{-\infty}^{\infty} \left[\frac{-\cos(\varphi)}{D} + \frac{4\sin^2(\varphi) - 3\cos^2(\varphi)}{D^2} + \frac{12\sin^2(\varphi)\cos(\varphi)}{D^3} - \frac{6\sin^4(\varphi)}{D^4}\right] d\epsilon. \tag{B.7}$$

## B.2 Resonant level model

The resonant level model has the scattering amplitude [23]

$$\Lambda(\epsilon, \varphi) = -2i\Gamma_\Sigma \epsilon^2 \sqrt{\epsilon^2 - 1} + (\epsilon_0^2 - \epsilon^2 + \Gamma_\Sigma^2)(\epsilon^2 - 1) + \frac{1}{2}(\Gamma_\Sigma^2 - \delta\Gamma^2)(1 - \cos(\varphi)), \tag{B.8}$$

where $\epsilon_0$ is a level offset and $\Gamma_\Sigma = \Gamma_L + \Gamma_R$ and $\delta\Gamma = \Gamma_L - \Gamma_R$. All quantities are defined in units of $\Delta$, and we focus on $\epsilon_0 = 0$ for simplicity for the numerical computations plotted in this manuscript. Then we also have

$$\bar{\Lambda}(i\epsilon) = -\frac{2F(\epsilon)}{\Gamma_\Sigma^2 - \delta\Gamma^2}, \quad \text{with} \tag{B.9}$$

$$F(\epsilon) = (\Gamma_\Sigma^2 + \epsilon_0^2)(1 + \epsilon^2) + 2\Gamma_\Sigma \epsilon^2 \sqrt{\epsilon^2 + 1} + \epsilon^2(1 + \epsilon^2). \tag{B.10}$$

We remark that the the first term dominates the strong coupling (short junction-like) limit and the last term dominates the weak coupling (dot-like) limit. When $\Gamma_\Sigma \sim 1$, all three terms contribute.

### B.3 Finite length model

The finite length model used here was introduced in [22]. In this model, the weak link has a single channel, a physical length $L$, and a Fermi velocity $v_F$. Then the effective coherence length is $\xi = \hbar v_F / \Delta$ and the dimensionless length scale is $l = L/\xi$. With this new parameter, we can construct a scattering model that produces the following scattering amplitude:

$$
\begin{aligned}
\Lambda_0(\epsilon) = &\frac{1}{2}\cos(2l\epsilon)(2\epsilon^2 - 1)(1 + R_L R_R) \\
&- i\sin(2l\epsilon)\epsilon\sqrt{\epsilon^2 - 1}(1 - R_L R_R) \\
&- \frac{1}{2}(R_L + R_R)\cos(2l\epsilon) \\
&+ 2\sqrt{R_L R_R}\cos(2lm)(\epsilon^2 - 1) \\
&- \frac{1}{2}(1 - R_L)(1 - R_R),
\end{aligned}
\tag{B.11}
$$

$$
\Lambda_\varphi(\varphi) = \frac{1}{2}(1 - R_L)(1 - R_R)(1 - \cos\varphi)
\tag{B.12}
$$

$$
\epsilon = \frac{E}{\Delta}
\tag{B.13}
$$

$$
m = \frac{\pi}{2l} + \frac{\epsilon_0}{\Delta},
\tag{B.14}
$$

where $R_{L,R}$ are the reflection coefficients at the left and right ends of the weak link region and $\epsilon_0$ is the level offset of lowest energy state in the normal region relative to the Fermi energy. For simplicity, we have focused on the case $\epsilon_0 = 0$ for the computations plotted in this manuscript. Finally,

$$
\bar{\Lambda}(i\epsilon) = -\frac{G(\epsilon)}{(1 - R_L)(1 - R_R)}, \quad \text{with}
\tag{B.15}
$$

$$
\begin{aligned}
G(\epsilon) = &(1 - R_L)(1 - R_R) + \cosh(2l\epsilon)(1 + R_L)(1 + R_R) + 2\epsilon^2\cosh(2l\epsilon)(1 + R_L R_R) \\
&+ 2\sinh(2l\epsilon)\epsilon\sqrt{1 + \epsilon^2}(1 - R_L R_R) + 4\cos(2lm)(1 + \epsilon^2)\sqrt{R_L R_R}.
\end{aligned}
\tag{B.16}
$$

### B.4 Logarithmic scale in dwell time

In Figure 6 here, the two models are compared in the same way as in Figure 1, but now with logarithmic scale on the lower axis. We remark in particular that the resonant level model shows substantial corrections to anharmonicity over several orders of magnitude in the coupling strength.

## C Analytic derivation of prefactors to nonanalytic dwell-time dependence

Here we sketch the derivation of the prefactors to the logarithmic dependence for small lengths and perfect transparency ($R_L = R_2 = \epsilon_0 = 0$). We find

$$
\bar{\Lambda}(i\epsilon) = -1 - (1 + 2\epsilon^2)\cosh(2l\epsilon) - 2\sinh(2l\epsilon)\epsilon\sqrt{1 + \epsilon^2},
\tag{C.1}
$$

and then for perturbatively small length we can approximate

$$
\bar{\Lambda}(i\epsilon) \approx -1 - (1 + 2\epsilon^2) - 4l\epsilon^2\sqrt{1 + \epsilon^2}.
\tag{C.2}
$$

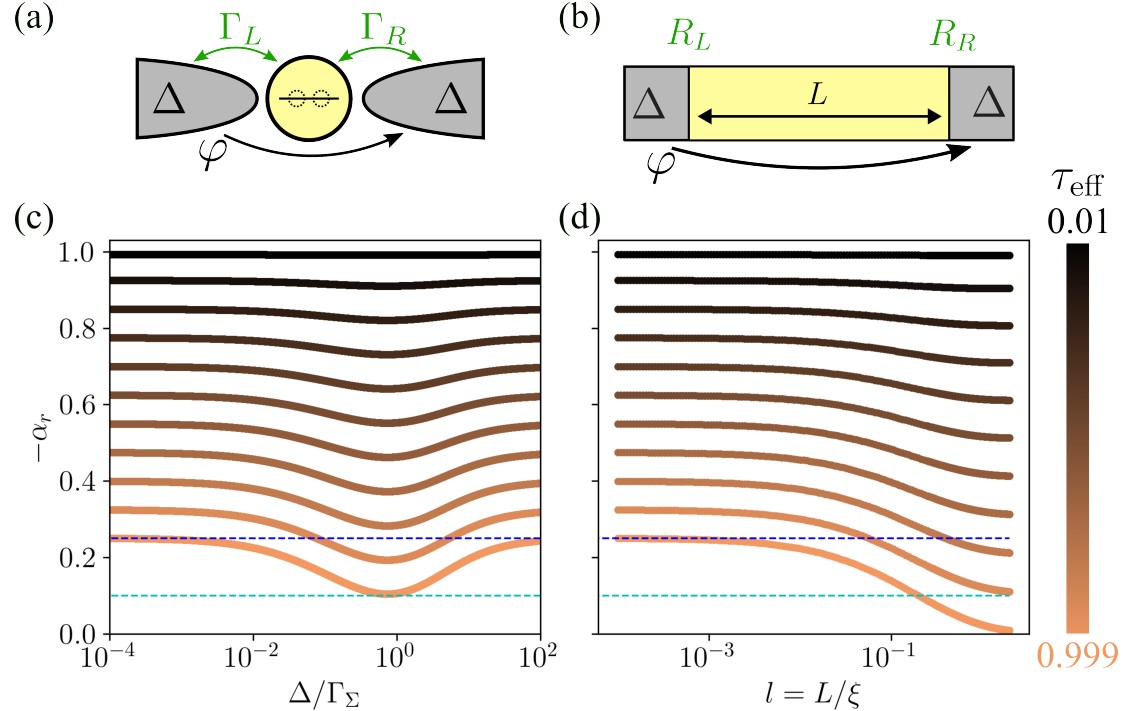

Figure 6: **Zero-impedance limit of transmon anharmonicity with logarithmic lower axis.** (a) Schematic of the short resonant level model (from [22]). (b) Schematic of the finite length single-channel model. (c-d) The ratio of the fourth and second order polynomial coefficients of the potential minimum for two models for evenly-spaced transparencies, with x-axis the parameter proportional to the dwell time in the circuit. (c) The resonant level model, as a function of the inverse total coupling strength for several different effective transparencies $\tau_{\text{eff}}$. (d) Finite length model, as a function of length for different $R_L = 1 - \tau_{\text{eff}}$, fixing $R_R = 0$. Blue dashed line is at 0.25 (lower limit of short junction formula), cyan dashed line is at 0.1 (lower limit of resonant level model).

For zero length, $l = 0$, we get the short junction result, e.g., for $c_2$:

$$c_2 = -\frac{1}{4\pi} \int_{-\infty}^{\infty} \frac{1}{\bar{\Lambda}(i\epsilon)} \, d\epsilon \tag{C.3}$$

$$= \frac{1}{8\pi} \int_{-\infty}^{\infty} \frac{1}{1 + \epsilon^2} \, d\epsilon \tag{C.4}$$

$$= \frac{1}{8}. \tag{C.5}$$

We can then calculate the correction to logarithmic precision:

$$c_2 = \frac{1}{8\pi} \int_{-\infty}^{\infty} \frac{1}{1 + \epsilon^2 + 2l\epsilon^2\sqrt{1 + \epsilon^2}} \, d\epsilon \tag{C.6}$$

$$\approx \frac{1}{8} - \frac{l}{2\pi} \int_{-\infty}^{\infty} \frac{\epsilon^2}{(1 + \epsilon^2)^{3/2}} \, d\epsilon \tag{C.7}$$

$$\approx \frac{1}{8} - \frac{l}{2\pi} \log \frac{1}{l}. \tag{C.8}$$

The last step is given by ending the integral at $1/l \gg 1$, as we cannot violate the assumption $\cosh 2l\epsilon \approx 1$ at this level of approximation. This procedure accurately produces the prefactor to the logarithm, but not the coefficient in the argument of the logarithm. Therefore, we obtain $a_2 = 1/2\pi$. A similar procedure on the integral for $c_4$ produces $a_4 = a_2/12$. The numerical calculations are consistent with these values for $a_2$ and $a_4$. Finally, we find $b_2 \approx 0.5615$ and $b_4 \approx 0.2066$ by comparison with numerics.

The resonant level model produces a similar logarithmic behavior as the coupling strength is reduced from infinity, i.e., replacing $l$ with $\Delta/\Gamma_\Sigma$, with the same values for $a_2$ and $a_4$. By comparing with numerics we find $b_2 \approx 0.446$ and $b_4 \approx 0.164$ for this model, distinct from (but close to) the values found for the finite length model.

# D   Indications of negative anharmonicity guarantee

## D.1   The considered single-channel models

At $\varphi = 0$, Equations B.6 and B.7 simplify:

$$c_2(0) = -\frac{1}{4\pi} \int_{-\infty}^{\infty} \frac{1}{\bar{\Lambda}(i\epsilon)} \, d\epsilon \,, \tag{D.1}$$

$$c_4(0) = \frac{1}{48\pi} \int_{-\infty}^{\infty} \left[ \frac{1}{\bar{\Lambda}(i\epsilon)} + \frac{3}{(\bar{\Lambda}(i\epsilon))^2} \right] d\epsilon \,, \tag{D.2}$$

which, when combined, give the equation used in the main text:

$$\alpha_r = -1 + 3 \frac{\int_{-\infty}^{\infty} |\bar{\Lambda}(i\epsilon)|^{-2} \, d\epsilon}{\int_{-\infty}^{\infty} |\bar{\Lambda}(i\epsilon)|^{-1} \, d\epsilon} \equiv -1 + \beta \,. \tag{D.3}$$

For our models, $\bar{\Lambda}(i\epsilon) \leq -2$, which guarantees $c_2 \geq 0$. Furthermore, we see that $\beta$ is a positive quantity, guaranteeing $\alpha_r > -1$.

The maximum possible correction is less obvious. For the resonant level model, the parameter $\beta$ is given by:

$$\beta = \frac{3}{2}(\Gamma_\Sigma^2 - \delta\Gamma^2) \frac{\int_0^{\infty} F(\epsilon)^{-2} d\epsilon}{\int_0^{\infty} F(\epsilon)^{-1} d\epsilon} \,. \tag{D.4}$$

The maximum value of $\beta$ is achieved when $\delta\Gamma = 0$ and $\epsilon_0 = 0$. Numerical evaluation of the integrals yields a maximum $\beta \approx 0.896$ at $\Gamma_\Sigma \approx 1.356$.

For the finite-length model, we have instead:

$$\beta = 3(1 - R_L)(1 - R_R) \frac{\int_0^{\infty} G(\epsilon)^{-2} d\epsilon}{\int_0^{\infty} G(\epsilon)^{-1} d\epsilon} \,. \tag{D.5}$$

We find that $\beta$ is a monotonously decreasing function of $R_L$ and $R_R$, and so its maximum value has to be found when $R_L = 0$ and $R_R = 0$. For these values, we find that $\beta$ increases monotonously with $l$. For $l \to \infty$, $\beta$ approaches an upper bound $\beta = 1$ which can be analytically obtained by replacing $G(\epsilon)$ with $1 + \cosh(2l\epsilon)$ inside the integrals. This limiting value reproduces the vanishing anharmonicity of an infinitely long ballistic junction.

## D.2   Random matrix approach

To further support this speculation, we checked random matrices satisfying the constraints for a multichannel SNS junction. We generate a normally-sampled random $N \times N$ real symmetric

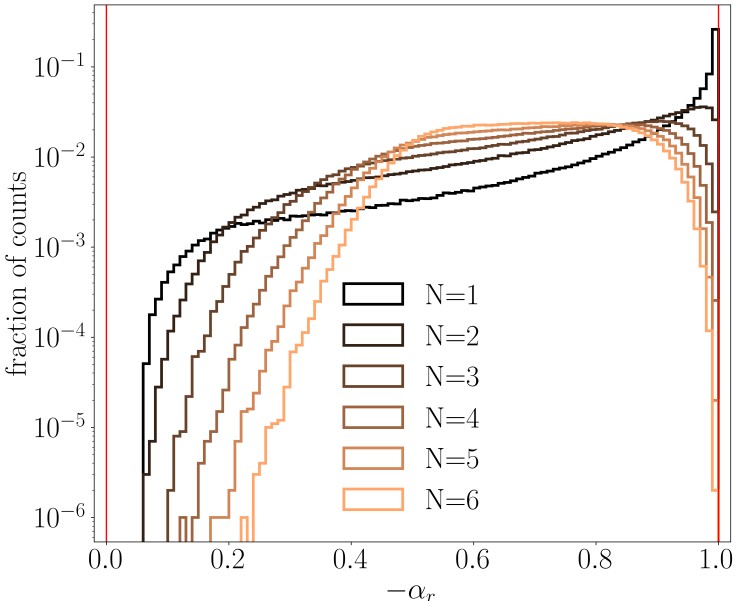

Figure 7: **Random matrix checks of SNS anharmonicity.** Normalized histogram of $-\alpha_r = -12c_4/c_2$ calculated from random matrices satisfying the constraints of a noninteracting SNS junction with time-reversal symmetry. The different curves are for different number of channels, ranging from 1 (black) to 6 (tan), with $10^6$ cases for each. Note that $c_2 > 0$ always resulted, in agreement with [38].

matrix $H_e$ to represent the normal states as well as a random positive diagonal $N \times N$ matrix $H_\Delta$ to represent superconducting pairing, and input those into the blocks below to create a random Bogoliubov-de Gennes Hamiltonian.

$$H_0 = \begin{pmatrix} H_e & H_\Delta \\ H_\Delta & -H_e \end{pmatrix}. \tag{D.6}$$

The number $N$ can be seen as representing the number of channels. The eigenvectors are then unfolded from the doubled degrees of freedom and used as the basis for block diagonal perturbation theory with perturbative phase fluctuations of order $n$ around $\langle \varphi \rangle = 0$:

$$\delta_n H = \frac{1}{n!} \begin{pmatrix} 0 & H'_\Delta(i)^n \\ H'_\Delta(-i)^n & 0 \end{pmatrix}, \tag{D.7}$$

where $H'_\Delta$ represents the superconductor on which the phase varies. We use the Pymablock package to solve this numerically [66,67]. We generated $10^6$ random BdG Hamiltonians each for $N = 1$ to 6 channels. The distributions for $\alpha_r$ are given in Figure 7, where we again find that the inequality $-1 \leq \alpha_r \leq 0$ holds. These cases did not include spin-orbit interaction – we also checked a million random Hamiltonians including spin-orbit interaction of 1, 2, and 3 channels, and found the inequality still to be respected.

We also considered behavior at $\varphi = \pi$. The scattering matrix at phase 0 and phase $\pi$ is different: at phase 0 it is positive semidefinite, whereas at phase $\pi$ it allows negative values. Indeed, at phase $\pi$ we find that both $c_4 > 0$ and $c_4 < 0$ cases are found within the random matrix approach.

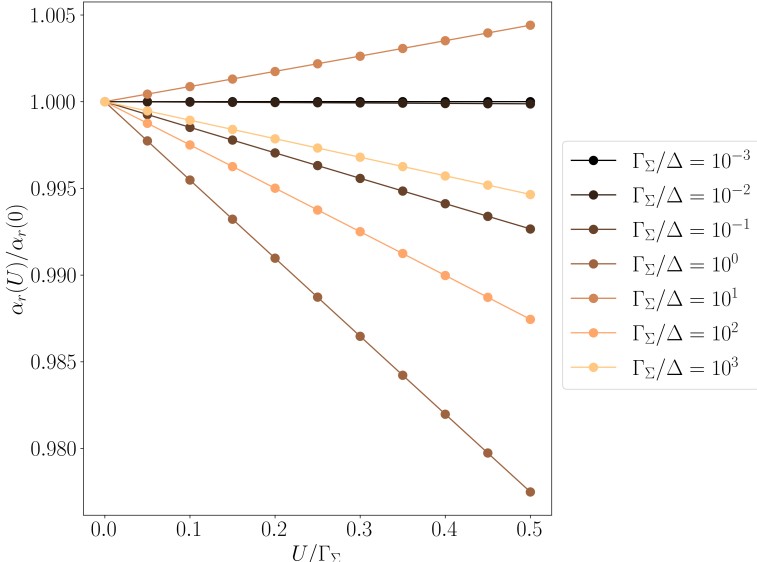

Figure 8: **Impact of charging energy on the resonant level.** The charging energy $U$ is varied in proportion to $\Gamma_\Sigma$ for 6 cases of $\Gamma_\Sigma$ from weak to strong coupling, using the perturbative-in-$U$ model of [21] The low-impedance anharmonicity is found be impacted at the few % level.

## E  Charging effects on anharmonicity for the resonant level

We use a previously developed model to inspect the effect of charging energy on the anharmonicity of a resonant level [21, 22]. We fix $\tau_{\text{eff}} = 0.9999$ and $\epsilon_0 = 0$. We vary $\Gamma_\Sigma/\Delta$ over 6 orders of magnitude from weak coupling to strong coupling, and for each coupling strength we vary the charging term $U$ from 0 to $\frac{1}{2}\Gamma_\Sigma$ (likely already violating the limits of the perturbation theory). The results are shown in Fig. 8, indicating that charging effects impact anharmonicity at the few % level. Notably, the relative impact is most strongly suppressed for weak coupling, consistent with the continuum hardly playing any role in the physics in that limit.

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
