# Peer review of "Nonlinearity of transparent SNS weak links decreases sharply with length"

_SciPost Physics, doi:SciPost Phys. 18, 091 (2025)_

## Round 2 · Referee Report · Jean-Damien Pillet (Referee 1) · 2024-12-29

Strengths

The manuscript exhibits remarkable strengths, such as:

1) Relevance for the field of gatemons and hybrid superconducting devices: The insights provided are highly applicable to ongoing research and technological development in this area.

2) Challenging preconceived notions: The study effectively debunks commonly held assumptions that can bias the analysis of experimental results.

3) Provision of analytical and approximate expressions: Key equations (e.g., 4, 7, 8, and 10) are presented clearly, offering valuable tools for the community.

4) Comprehensive and insightful literature review: The bibliography is thorough, including both foundational and recent references, and introduces several noteworthy papers that are useful to the field.

5) Comparison with recent experimental results: The manuscript addresses contemporary findings, including the intriguing case of the wide junction, which enhances its practical relevance.

6) Experimental strategies for identifying operational regimes: Practical methods, such as analyzing higher-order anharmonicities, are proposed, with a helpful appendix showcasing an application to existing data.

Weaknesses

1) Limited discussion beyond the single-channel one-dimensional case: While Figure 7 provides some insight, the focus remains predominantly on this scenario. This is, of course, the most illustrative and relevant case, but it is common to encounter multiple conduction channels, whose influence would benefit from further exploration.

2) Minimal discussion of interaction effects: The role of interactions is barely addressed. Since this is not the primary focus of the article, it represents a minor limitation.

3) Transition between the quantum dot and finite-length wire regimes: The comparison between these two cases is highly engaging but occasionally difficult to follow. This slightly affects the readability of the article, although the figures provide excellent clarity on this point.

4) Relevance of dwell time in the quantum dot regime: In the quantum dot case, the notion of dwell time is emphasized. However, it is not entirely clear that this is the most pertinent parameter, and further clarification would be helpful.

Report

This theoretical article focuses on the nonlinearity introduced by an SNS Josephson junction, where the weak link (N) is a normal wire (i.e., non-superconducting and without any particular internal properties), in superconducting circuits. This nonlinearity is highly beneficial for various applications, such as creating two-level systems for qubits, as well as other devices like quantum-limited amplifiers and frequency converters.

The article is authored by established specialists in the field who ask pertinent questions and, in particular, explore the impact of the length $L$ of the weak link on the nonlinearity $\alpha$ introduced by the SNS junction. I find this work extremely interesting and genuinely useful for the community working on hybrid superconducting circuits, where the weak link is typically implemented using low-dimensional quantum conductors, such as InAs nanowires or graphene. One of the paper’s greatest strengths (as aptly highlighted by its title) is its demonstration that the length $L$ can have a dramatic influence when it becomes comparable to $\xi$, the superconducting coherence length. This influence can result in a reduction of $\alpha$ of over an order of magnitude compared to a more standard Josephson tunnel junction, such as those used in transmon design. This finding is particularly striking and challenges the conventional notion that this parameter should have only a minor effect when implementing a gatemon, the hybrid version of the transmon. The result is clearly presented in Figure 1d and is further elaborated upon in the rest of the article.

The manuscript also exhibits other remarkable qualities, such as providing highly useful analytical expressions for the community, including equations 4, 7, 8, and 10. To my knowledge, these expressions are not explicitly detailed (at least in this form) in any other references, making this paper a true compact reference with extremely useful tools for analyzing experimental data obtained from gatemons. Another aspect I particularly appreciated is the meticulously curated bibliography, which is both comprehensive and relevant, including both older foundational references and more recent works. Thanks to the authors, I discovered several highly useful articles I was previously unaware of.

From a more pragmatic perspective, I believe this manuscript offers significant added value by proposing methods to identify the operational regime of a gatemon-type qubit, such as by comparing higher-order anharmonicities. Furthermore, the authors include an example analysis in the appendix, using data already present in the literature. This is particularly illuminating for understanding the procedure to follow.

In summary, I greatly appreciate this article and fully support its publication in SciPost without reservation.

Since the refereeing process requires me to identify potential weaknesses of this article, I have listed these in the dedicated section, which I hope might assist the authors in improving the readability of the manuscript.

I would like to add a few comments here, which are more aimed at satisfying my curiosity (and perhaps that of other readers!) than at suggesting actual corrections.

1) The article shows in Figure 4 that nonlinearities rapidly converge to saturation as the ratio $f_{01}/E_C$, and therefore $E_J/E_C$, tends towards large values. Strikingly, this convergence is much faster for the short and long junction cases than for the tunnel junction case. I would have been very interested to know whether this is related to the fact that hybrid qubits reach the transmon regime faster than tunnel junctions as $E_J/E_C$ increases - that is, their charge dispersion flattens much more quickly. While this point has been discussed elsewhere, particularly in the early references cited by the authors, I believe the present work could provide a deeper perspective on this question. Do the authors have any insight on this point ?

2) The authors introduce the concept of dwell time in the quantum dot case, defined as $\Gamma/\Delta$. This definition aligns well with intuition, as $h/\Gamma$ indeed gives the order of magnitude of the time an electron spends in a quantum dot. The normalization by $\Delta$ is reasonable, as it represents another relevant energy scale in the problem, and it allows for a dimensionless quantity to plot the nonlinearity (Figure 1c).

However, the parallel with $L/\xi$, used for the long junction case, seems harder to grasp. This dimensionless quantity makes sense as it corresponds to the ratio of the Thouless energy to the superconducting gap and thus indicates how many box energy levels fit within $\Delta$, directly giving the number of Andreev states. I fail to see the equivalent interpretation for the dwell time $\Gamma/\Delta$ (as the authors clearly state it, it does not change the number of Andreev Bound States).

Moreover, these parameters do not appear to have the same influence on $\alpha$, as the latter finally increases with the dwell time $\Delta/\Gamma$ in the quantum dot case, while it converges to zero with $L/\xi$ in the long junction case. It is not clear to me that these two parameters play comparable role and I don't fully agree with the statement of the author " the dwell time can be used as a common parameter for comparisons to a junction of finite length."

In my view, the ratio $\Delta/\Gamma$ primarily controls the contribution of the continuum to the Josephson current. It seems that the greater the continuum contribution (essentially the case for $\Delta/\Gamma \sim 1$), the weaker the nonlinearity. This contribution vanishes in the limits $\Gamma \ll \Delta$ and $\Gamma \gg \Delta$, which could explain the minimum in between observed by the authors. This interpretation is also consistent with Figure 1d, where larger $L/\xi$ values correspond to a relatively larger contribution from the continuum (compared to Andreev Bound States). For instance, see Figure 6 of Phys. Rev. B 62, 1319 (2000). Why a strong continuum contribution to the supercurrent minimizes $\alpha$ remains unclear, but I would have been very interested to hear the authors' perspective on this.

To conclude this report, I reiterate my support for the publication of this manuscript in SciPost.

I have found a small number of typos that I have detailes in the "requested changes" section.

Requested changes

1) On page 4, the authors wrote $\Gamma_\Sigma \ll 0$. I assume they meant $\Gamma_\Sigma \ll \Delta$.

2) In Fig. 4, the yellow curve corresponding to $\alpha_{345}$ is missing from the legend.

3) In Fig. 1, the y-axis is labeled $2c_1/\Delta$. This does not seem correct, as $c_1$ should be a constant. I suspect the authors meant $dU/d\varphi \times 1/\Delta$ (or something proportional).

4) Figure 3(d) is referenced twice but does not exist.

Recommendation

Publish (easily meets expectations and criteria for this Journal; among top 50%)

  • validity: top
  • significance: high
  • originality: high
  • clarity: good
  • formatting: excellent
  • grammar: perfect

Author:  Valla Fatemi  on 2025-02-04  [id 5182]

(in reply to Report 1 by Jean-Damien Pillet on 2024-12-29)
Category:
answer to question
correction

Thank you for the positive and complimentary review. We have corrected the errors and typos in the "requested changes" section. Regarding your comments:

  1. We agree that the anharmonicities converge less quickly with tunneling junctions than in the other two cases: for tunnel, short transparent, and long transparent junctions, the energy differences (E(\pi)-E(0)=2, 4, \pi^2) for equivalent (c_2) at zero phase. This barrier height is key to determining the tunneling rate of the phase particle between minima -- for EPRs which naturally have higher barriers, the tunneling rate will be more quickly suppressed as the impedance is lowered (going towards the right of the figure). This tunneling rate of the phase particle should determine the charge dispersion. So, there may indeed be a relationship between the anharmonicity and the charge dispersion, although we acknowledge that the barrier height will depend on all (c_n), and we are not sure yet that this should be totally generic for all SNS.

  2. For a simple model assuming a linearized dispersion, (\xi \propto v_F/\Delta), where (v_F) is the Fermi velocity. Therefore, the ratio (L/\xi \rightarrow (L/v_F)/\Delta), and we can identify (L/v_F) as the time of flight from S to S within the normal region. In this way the concept of Thouless energy (which people often associate with diffusive transport only) is naturally extended to the ballistic system and to the quantum dot; we can see Thouless energy as the inverse dwell time. Therefore, we promote this ratio as also being a dwell time in the normal region. We have added a sentence to the main text with the above formulation to explain this in this way, which we believe should help clarify this relationship. Moreover, with this definition, the first order corrections due to dwell time are mathematically identical in structure and in half of the coefficients (see equation 7 and surrounding discussion), so we think that the dwell time can be a unifying concept. For long dwell times (or higher order corrections), the two models indeed result in different effects.

  3. We unfortunately do not have for the referee a comprehensive intuitive picture for why the continuum contribution suppresses anharmonicity relative to the linear response. We can note one example in the quantum dot case in the limit of weak coupling, which uses results from [10.1103/PhysRevLett.129.227701]. The continuum contribution will be largely cosine-like with a (\pi) phase shift, while the dot-like Andreev level would disperse as the short junction formula. The short junction formula alone can have (-1 < \alpha_r < -0.25), and a (\pi)-shifted cosine alone at zero phase would have (\alpha_r = +1). So, in this case, the continuum contribute relatively more 'opposing' anharmonicity than it does linear inverse inductance. A similar logic may hold in the other regimes.

  4. We also remark on a few of the weaknesses mentioned:

    1. We agree the multichannel case is important for experiments. However, we believe that it is beyond the scope of our already somewhat long paper, in which we desired to focus on the most qualitatively important features of dwell time effects. The treatment of parallel noninteracting channels is sufficiently straightforward in principle, and we believe it best suited for a future work, perhaps in concert with experiments.
    2. We found that weak charging effects do not have much impact, as shown in in Appendix E and Figure 8, so we did not pursue this thread further. Going to limits with stronger interactions would require more sophisticated models, which we believe is best for future work by researchers specializing in those models.
    3. We have added remarks to help sustain the connection between the two cases on pages 2 and 6.
    4. We are not sure what exactly the referee is requesting clarity on for the dot case. There are basically two parameters, the effective transparency (related to the imbalance of the tunneling rates) and the total tunneling rate (which we connect to the dwell time), and both of these are treated in the paper.

---

## Round 2 · Referee Report · Anonymous (Referee 2) · 2025-1-12

Strengths

1) The subject of the paper is very relevant for the field of qubits in superconducting circuit systems, particularly for transmon (or gatemon) qubits, a very active field in the community.

2) Identification of a simple yet crucial factor that may be playing a role in experiments and that is typically overlooked: finite-length effects of shortish (but not zero-length) SNS junctions.

2) Since the main result of this work is precisely a lower anharmonicity in shortish, transparent SNS junctions, the authors provide at the end a "prospective usefulness of low anharmonicity junctions" that I consider enlightening and motivating.

3) Apart from its usefulness for practical applications and experiments, I think these results are of importance for the theoretical understanding and description of SNS junctions. Particularly, for non-interacting, single level SNS junctions of "intermediate" length.

4) Derivation of several analytical results, discussion of different limits, comparison between two important basic models for single-channel SNS junctions, varios appendixes with full model, explanations and derivation details, comprehensive review of the literature and state of the art in the field, comparison with experiments.

Weaknesses

I cannot really find important weaknesses. This is a theoretical work based on basic, phenomenological models (as opposed to microscopic), and thus aims at providing simple, general, ideal results and behaviors rather than describing in detail particular experiments. Thus, it probably overlooks the effects of some other realistic factors that may be playing a role in some experiments (multichannel effects, disorder, interactions, self-consistency...). But as I say, I don't think that is the aim of this paper or a shortcoming for the validity and usefulness of this work.

Report

In this work the authors identify a common simplifying assumption that is done when describing SNS junctions in the context of microwave circuits, and that turns out not to be correct in realistic experiments. For "shortish" SNS junctions, those with L≲ξ, the short junction limit (L≪ξ) is typically used. However, they uncover that a finite length of the weak link strongly suppresses the junction's nonlinearity (and thus its anharmonicity) compared to its zero-length limit; the reason being that the continuum contribution to the phase-dependent part of the ground-state energy starts to play a role.
To prove this fact and to provide several analytical derivations, the authors inspect two basic models for non-interacting single-channel SNS junctions (a short resonant level model and a ballistic finite-length model), finding similar (at least related) results. They find that the suppression of the anharmonicity can be up to a factor of ten (compared to the zero-length limit) whereas the critical current decays only by a factor of two.

I think this work is very relevant for the current investigations in microwave circuits, transmon qubits and related qubits. Perhaps it is even more relevant for other applications where a low anharmonicity is useful, as the authors point out in Section 4.4. Thus, I think this work is very timely. Apart from the comments I have written in the "Strengths section", I think that the paper is very well written, the calculations look sound, and the results are well explained, justified and discussed.

For these reasons, I think this manuscript meets the Journal's acceptance criteria and I thus recommend it for publication in SciPost Physics.

I just have a question and a comment:

Q: Do I understand correctly that the results the authors derive correspond to a (single-channel) SNS junction where the Andreev approximation (Δ≪μ) is necessary? Or could they be applied to junctions where μ∼Δ, like the hybrid superconductor-semiconductor heterostructures used e.g. in gatemons?

C: In page 8, the authors say: "In Figure 4(a), we show calculations of the first four anharmonicities as a function of the scale of the potential (a multiplicative factor)...". I don't understand what they mean by "the scale of the potential (a multiplicative factor)".

Requested changes

I have found some typos:

  • In page 4: "while the limits ΓΣ ≫∆ and ΓΣ ≪0 reproduce the physics of Eq. (1)". It should be ΓΣ ≪∆.

  • In pages 10 and 11: I think the authors are referring to Fig. 1(d) (?) instead of Fig. 3(d).

Recommendation

Publish (easily meets expectations and criteria for this Journal; among top 50%)

  • validity: high
  • significance: high
  • originality: good
  • clarity: high
  • formatting: excellent
  • grammar: perfect

Author:  Valla Fatemi  on 2025-02-04  [id 5183]

(in reply to Report 2 on 2025-01-12)
Category:
answer to question
correction

Thank you for the positive and complimentary review. We have corrected the errors and typos in the "requested changes" section. Regarding your comments:

  1. It is correct that the long-junction results are derived in the context of linearized dispersion relation at the chemical potential, so that the Andreev approximation applies. Violation of the Andreev approximation can be interpreted as energy-dependent normal scattering at the N-S interfaces, and so it could be included and accounted for. The effective transparencies that result may not be identical to the effective transparencies identified here, and it would be a useful point for future work to consider addressing. That said, the generic result in equation (10), (-1 \leq \alpha_r \leq 0), was found also with a random matrix theory approach, which does not make any such approximation.

  2. What we mean is that we fix the shape of the energy-phase relation and multiply it by a constant to achieve the dependence along the x-axis. We have modified the text to achieve better clarity on this point. Thank you for mentioning the awkwardness of the phrasing.

---

## Round 4 · Referee Report · Jean-Damien Pillet (Referee 1) · 2025-2-5

Report

The responses and corrections provided by Valla Fatemi and the coauthors are appropriate and insightful. I appreciate their effort and time in addressing my questions.

I have no further comments and reaffirm my strong support for the publication of this work in SciPost.

Recommendation

Publish (easily meets expectations and criteria for this Journal; among top 50%)

---

## Round 4 · Referee Report · Anonymous (Referee 2) · 2025-2-12

Report

The authors have answered satisfactorily all my comments/questions. I think that the manuscript is now ready for publication.

Recommendation

Publish (easily meets expectations and criteria for this Journal; among top 50%)

---

## Round 4 · Author Response

Dear Editor,

We have corrected the errors and typos in the ``requested changes" sections of the referee reports, and also made adjustments for clarity and added information based on the questions, comments, and noted weaknesses. We resubmit this improved version of the manuscript for consideration for publication.

Best regards,

Valla Fatemi, on behalf of the co-authors.

---

## Round 4 · List of Changes

All the requested changes by the referees were made.

We added a remark about how to think about dwell time in the ballistic, nonzero-length junction.

Pages 2 and 6 have some added text to better maintain the connection between the two models.

We added text to clarify the meaning of the multiplicative factor used in Figure 4.

---

## Editorial Decision

published